# The Pessimistic Limits and Possibilities of Margin-based Losses in Semi-supervised Learning

**Jesse H. Krijthe**
Radboud University, The Netherlands
jkrijthe@gmail.com

**Marco Loog**
Delft University of Technology, The Netherlands
University of Copenhagen, Denmark
m.loog@tudelft.nl

## Abstract

Consider a classification problem where we have both labeled and unlabeled data available. We show that for linear classifiers defined by convex margin-based surrogate losses that are decreasing, it is impossible to construct *any* semi-supervised approach that is able to guarantee an improvement over the supervised classifier measured by this surrogate loss on the labeled and unlabeled data. For convex margin-based loss functions that also increase, we demonstrate safe improvements *are* possible.

## 1 Introduction

Semi-supervised learning has been reported to deliver encouraging results in various settings, e.g. for object detection in computer vision (Rasmus et al., 2015), protein function prediction from sequence data (Weston et al., 2005) or prediction of cancer recurrence (Shi & Zhang, 2011) in the bio-medical domain and part-of-speech tagging in natural language processing (Elworthy, 1994). In other settings, however, using unlabeled data has been shown to lead to a decrease in performance when compared to the supervised solution (Elworthy, 1994; Cozman & Cohen, 2006). For semi-supervised classifiers to be used safely in practice, we may at least want some guarantee that they do not reduce performance compared to their supervised alternatives. Some have attempted to provide such guarantees either empirically by restrictions on the parameters to be estimated (Loog, 2010) or under particular assumptions on the data (Li & Zhou, 2015). In general, however, it is unclear for what classifiers one can construct 'safe' semi-supervised approaches that can be expected to not decrease performance, or whether this is at all possible.

### 1.1 Safety and Pessimism

This work explores whether and, if so, how we can guarantee unlabeled data to improve, or at least not decrease the performance of a semi-supervised classifier in comparison to a supervised classifier. 'Pessimism' refers to the property that we want this guarantee to hold for every single instantiation of a problem, even for the worst possible unknown labeling of the unlabeled data. The reason we choose such a strict criterion is that it is the only criterion that can guarantee (with probability one), that performance degradation will not occur, for the particular dataset one is faced with. Therefore, a semi-supervised approach can only be called truly safe if it guarantees non-degradation of performance in this pessimistic sense. Note that the labelings that we will be considering are not as pessimistic as they first appear: because performance is compared to the supervised classifier, these labelings will be optimistic with respect to the supervised classifier, as will become apparent when we formally define the criterion for safe semi-supervised learning in Equation (3).

We compare the performance of the supervised and semi-supervised classifier measured on the labeled and unlabeled data. This is, strictly speaking, a transductive setting (Joachims, 1999), where one

measures performance on a specifically defined set of objects, and not a semi-supervised setting where one measures performance on unseen objects generated from the same distribution as the training data. There are two reasons this transductive setting is interesting in the context of safe semi-supervised learning. 1. The performance criterion in this setting corresponds to the performance criterion we would observe and optimize for if the labels for all objects would be available. 2. As the number of unlabeled objects grows, and they start to better represent the distribution of interest in the inductive/semi-supervised setting, the limits and possibilities that we derive continue to hold, while we converge to the setting where the marginal distribution of the inputs is assumed to be known that is considered in other work on the possibilities of semi-supervised learning (Sokolovska et al., 2008; Ben-David et al., 2008). All in all, while we consider a transductive setting, it should be clear that our analysis does provide valid insights into the safe semi-supervised setting as well.

## 1.2 The Use of Surrogate Losses

Important in this work, is that we take the view that a semi-supervised version of, for instance, logistic regression is a classifier that still attempts to minimize logistic loss, but uses unlabeled data to improve its ability to do so. So it should be judged on how well it generalizes in terms of this intrinsic loss. If we were to compare performance in terms of some other loss, like the error rate, one runs the risk of attributing improvements to the use of unlabeled data that are, in fact, caused by other changes to the classifier. For instance, the semi-supervised classifier might implicitly use some other surrogate loss that turns out to be better aligned with the loss used for evaluation.

Therefore, as our definition of performance, we consider the surrogate loss the classifier optimizes and compare this loss for the supervised and the semi-supervised learner on the combined labeled and unlabeled data. The surrogate loss corresponds to the loss one would minimize if we did have labels for the unlabeled objects. Considering the same criterion in the supervised and semi-supervised case aligns the goal of constructing a semi-supervised classifier with the one used when constructing a supervised classifier. It avoids conflating improved performance based on a change in surrogate loss function with improvements gained by the availability of unlabeled data. For the same reason we also keep the regularization parameter fixed in the objective functions of the supervised and semi-supervised classifiers.

## 1.3 Outline

The main conclusion from our analysis (Theorems 1 and 2) is that for classifiers defined by convex margin-based surrogate losses that are decreasing, it is impossible to come up with *any* semi-supervised approach that is able to guarantee safe improvement. We also consider the case of losses that are not decreasing and in particular study the quadratic loss. We show under what conditions it *is* possible in this case to come up with a semi-supervised classifier that provides safe improvements over the supervised classifier.

The rest of this work is structured as follows. We start by introducing margin-based loss functions in the empirical risk minimization framework and the extension to the semi-supervised setting. In this, we only treat binary linear classifiers. Though not a real restriction, it does simplify our exposition and allows us to focus on the core ideas. In Section 3, we formalize our strict notion of safe semi-supervised learning. We first show that for the class of decreasing loss functions it is impossible to derive any semi-supervised learning strategy that is not worse than the supervised classifier for all possible labelings of the unlabeled data. We then consider the case of soft assignment of unlabeled objects to classes. Here, too, it is impossible to provide a strict improvement guarantee for this class of loss functions. We subsequently show for what losses it is possible to get strict improvements. In Section 5 we apply the theory to a few well-known loss functions. In Section 6 we discuss how these results relate to other results on the (im)possibility of (safe) semi-supervised learning and what the implications of these results are for other safe approaches.

## 2 Preliminaries

We consider binary linear classifiers in the empirical risk minimization framework. Let $\mathbf{X}$ be an $L \times d$ design matrix of $L$ labeled objects, where each row $\mathbf{x}^\top$ is a $d$-dimensional vector of feature values corresponding to each labeled object. Let $\mathbf{y} \in \{-1, +1\}^L$ be the corresponding label vector.

The vector $\mathbf{w} \in \mathbb{R}^d$ contains the weights defining a linear classifier through $\text{sign}(\mathbf{x}^\top \mathbf{w})$. We consider convex margin-based surrogate loss functions, which are loss functions of the form $\phi(y\mathbf{x}^\top \mathbf{w})$. Many well-known classifiers can be described in this way, including support vector machines, least squares classification, least squares support vector machines and logistic regression (Bartlett et al., 2006).

## 2.1 Empirical Risk Minimization

In the empirical risk minimization framework a classifier is obtained by minimizing a chosen surrogate loss $\phi$ over a set of training objects plus an optional regularization term $\Omega$, which we take to be a convex function of $\mathbf{w}$:

$$R_\phi(\mathbf{w}, \mathbf{X}, \mathbf{y}) = \sum_{i=1}^{L} \phi(y_i \mathbf{x}_i^\top \mathbf{w}) + \lambda \Omega(\mathbf{w}) \,. \tag{1}$$

By minimizing this with respect to $\mathbf{w}$ we get a supervised classifier:

$$\mathbf{w}_{\text{sup}} = \arg\min_{\mathbf{w}} R_\phi(\mathbf{w}, \mathbf{X}, \mathbf{y}) \,.$$

In the semi-supervised setting, we have an additional design matrix corresponding to unlabeled objects $\mathbf{X}_\text{u}$, sized $U \times d$, with unknown labels $\mathbf{y}_\text{u} \in \{-1, +1\}^U$. We therefore consider the corresponding semi-supervised risk function:

$$R_\phi^{\text{semi}}(\mathbf{w}, \mathbf{X}, \mathbf{y}, \mathbf{X}_\text{u}, \mathbf{q}) = R_\phi(\mathbf{w}, \mathbf{X}, \mathbf{y}) + \sum_{i=1}^{U} q_i \phi(\mathbf{x}_i^\top \mathbf{w}) + (1 - q_i)\phi(-\mathbf{x}_i^\top \mathbf{w}) \,, \tag{2}$$

where $\mathbf{q} \in [0, 1]^U$ are what we will refer to as *responsibilities*, indicating the unknown and possibly 'soft' membership of each object to a class. For instance, if the true labels were known these would correspond to 'hard' responsibilities $\mathbf{q}^{\text{true}} \in \{0, 1\}^U$ and the semi-supervised risk formulation becomes equal to the supervised risk formulation in Equation (1), where the sum is now over the $L$ labeled objects and the $U$ objects for which we did not have a label.

## 3 Limits of Safe Semi-supervision

Even though we know the true labeling of the unlabeled objects in Equation (2) belongs to some $\mathbf{q} \in \{0, 1\}^U$, we do not know which one. Now, a semi-supervised procedure $\mathbf{w}_{\text{semi}}$ is *safe* if it is guaranteed to attain a loss on the labeled and unlabeled objects equal to or lower than the supervised solution for all possible labelings of the data, since this is guaranteed to include the true labeling of the unlabeled objects. We first formalize this definition of safety, then consider the cases of hard and soft labeling, and come to our negative results: for many loss functions safe semi-supervision is, in fact, not possible. Positive results follow in Section 4.

### 3.1 Hard labeling

Let $D_\phi$ denote the difference in terms of the chosen loss $\phi$ on a set of objects between a new classifier $\mathbf{w}$ and the supervised classifier $\mathbf{w}_{\text{sup}}$ for some set of responsibilities for the unlabeled data:

$$D_\phi(\mathbf{w}, \mathbf{w}_{\text{sup}}, \mathbf{X}, \mathbf{y}, \mathbf{X}_\text{u}, \mathbf{q}) = R_\phi^{\text{semi}}(\mathbf{w}, \mathbf{X}, \mathbf{y}, \mathbf{X}_\text{u}, \mathbf{q}) - R_\phi^{\text{semi}}(\mathbf{w}_{\text{sup}}, \mathbf{X}, \mathbf{y}, \mathbf{X}_\text{u}, \mathbf{q}) \,.$$

The true unknown labels can, in principle, correspond to any $\mathbf{q} \in \{0, 1\}^U$. For a semi-supervised classifier $\mathbf{w}_{\text{semi}}$ to be *safe* we therefore need that:

$$\max_{\mathbf{q} \in \{0,1\}^U} D_\phi(\mathbf{w}_{\text{semi}}, \mathbf{w}_{\text{sup}}, \mathbf{X}, \mathbf{y}, \mathbf{X}_\text{u}, \mathbf{q}) \leq 0 \,. \tag{3}$$

If the inequality is strict for at least one instantiation of $\mathbf{q}$, the semi-supervised solution is different from the supervised solution and potentially better. Is it possible to construct some semi-supervised strategy that has this guaranteed improvement over the supervised solution for margin-based surrogate losses? The following theorem gives a condition under which this strict improvement is never possible.

**Theorem 1.** *Let $\mathbf{w}_{\text{sup}}$ be a minimizer of $R_\phi(\mathbf{w}, \mathbf{X}, \mathbf{y})$ and assume it is unique. If $\phi$ is a decreasing margin-based loss function, meaning $\phi(a) \geq \phi(b)$ for $a \leq b$, then there is no safe semi-supervised procedure which guarantees Equation (3) while having at least one $\mathbf{q}^* \in \{0, 1\}^U$ for which $D_\phi(\mathbf{w}_{\text{semi}}, \mathbf{w}_{\text{sup}}, \mathbf{X}, \mathbf{y}, \mathbf{X}_\text{u}, \mathbf{q}^*) < 0$.*

*Proof.* We are going to prove this by contradiction. Assume $D_\phi(\mathbf{w}_{\text{semi}}, \mathbf{w}_{\text{sup}}, \mathbf{X}, \mathbf{y}, \mathbf{X}_{\text{u}}, \mathbf{q}^*) < 0$ and define $M$ to be $R_\phi(\mathbf{w}_{\text{semi}}, \mathbf{X}, \mathbf{y}) - R_\phi(\mathbf{w}_{\text{sup}}, \mathbf{X}, \mathbf{y})$. The latter is the difference in the supervised objective function between the semi-supervised and supervised classifier. Based on our assumption we can now write

$$
\begin{aligned}
M + \sum_{i=1}^{U} q_i^* (\phi(\mathbf{x}_i^\top \mathbf{w}_{\text{semi}}) - \phi(\mathbf{x}_i^\top \mathbf{w}_{\text{sup}})) & \quad\quad\quad (4) \\
+ (1 - q_i^*)(\phi(-\mathbf{x}_i^\top \mathbf{w}_{\text{semi}}) - \phi(-\mathbf{x}_i^\top \mathbf{w}_{\text{sup}})) & \quad < 0 \, .
\end{aligned}
$$

Let $A_i = \phi(\mathbf{x}_i^\top \mathbf{w}_{\text{semi}}) - \phi(\mathbf{x}_i^\top \mathbf{w}_{\text{sup}})$ and $B_i = \phi(-\mathbf{x}_i^\top \mathbf{w}_{\text{semi}}) - \phi(-\mathbf{x}_i^\top \mathbf{w}_{\text{sup}})$. Since $\phi$ is decreasing, either $A_i \geq 0$ and $B_i \leq 0$, or $A_i \leq 0$ and $B_i \geq 0$. Set $q_i^{\text{new}} = 1$ in the former case and $q_i^{\text{new}} = 0$ in the latter. Then, when using $\mathbf{q}^{\text{new}}$ instead of $\mathbf{q}^*$ in Equation (4), the sum will be non-negative. Also, $M > 0$, because $\mathbf{w}_{\text{sup}}$ is the unique minimizer of $R_\phi(\mathbf{w}, \mathbf{X}, \mathbf{y})$ and $\mathbf{w}_{\text{semi}} \neq \mathbf{w}_{\text{sup}}$. We therefore have that

$$
D_\phi(\mathbf{w}_{\text{semi}}, \mathbf{w}_{\text{sup}}, \mathbf{X}, \mathbf{y}, \mathbf{X}_{\text{u}}, \mathbf{q}^{\text{new}}) > 0 \, .
$$

which contradicts Equation (3). $\qquad\square$

**Remark 1.** *Alternatively, we can drop the requirement that $\mathbf{w}_{\text{sup}}$ is the unique minimizer of $R_\phi(\mathbf{w}, \mathbf{X}, \mathbf{y})$ by requiring the loss functions to be strictly decreasing.*

### 3.2 Beyond Hard Labelings

In Equation (3) we considered improvement over all hard labelings of the unlabeled data. Alternatively we could also consider improvements for the larger set of all soft assignments of objects to classes, defining safety to mean

$$
\max_{\mathbf{q} \in [0,1]^U} D_\phi(\mathbf{w}_{\text{semi}}, \mathbf{w}_{\text{sup}}, \mathbf{X}, \mathbf{y}, \mathbf{X}_{\text{u}}, \mathbf{q}) \leq 0 \, . \quad\quad (5)
$$

If there is at least one $\mathbf{q} \in [0,1]^U$ for which the inequality is strict, the semi-supervised solution is potentially better than the supervised solution. There are several reasons why this is an interesting relaxation to consider. First of all, it requires the semi-supervised solution to guarantee improvements for a larger class of responsibilities than just the hard labelings, meaning it becomes more difficult to construct a procedure with this property. If a procedure guarantees improvement in this sense, it implies it also works for all possible hard labelings. Secondly, it corresponds to a scenario different from the hard labeling where there is uncertainty in the labels of the unlabeled objects. And lastly, the convex constraint makes the problem more amenable to analysis.

The set of classifiers induced by all different responsibilities turns out to be a useful concept in the remainder of this paper.

**Definition 1.** *The constraint set $\mathcal{C}_\phi$ is the set of all possible classifiers that can be obtained by minimizing the semi-supervised loss for any vector of responsibilities $\mathbf{q}$ assigned to the unlabeled data, i.e.,*

$$
\mathcal{C}_\phi = \left\{ \arg\min_{\mathbf{w}} R_\phi^{\text{semi}}(\mathbf{w}, \mathbf{X}, \mathbf{y}, \mathbf{X}_{\text{u}}, \mathbf{q}) \Big| \mathbf{q} \in [0,1]^U \right\} \, .
$$

The following lemma provides an intermediary step towards our second negative result. It tells us that no strict improvement is possible if the supervised solution is already part of the constraint set.

**Lemma 1.** *If $R_\phi(\mathbf{w}, \mathbf{X}, \mathbf{y})$ is strictly convex and $\mathbf{w}_{\text{sup}} \in \mathcal{C}_\phi$, then there is a soft assignment $\mathbf{q}^*$ such that for every choice of semi-supervised classifier $\mathbf{w}_{\text{semi}} \neq \mathbf{w}_{\text{sup}}$, $D_\phi(\mathbf{w}_{\text{semi}}, \mathbf{w}_{\text{sup}}, \mathbf{X}, \mathbf{y}, \mathbf{X}_{\text{u}}, \mathbf{q}^*) > 0$.*

*Proof.* As $\mathbf{w}_{\text{sup}} \in \mathcal{C}_\phi$ there is a soft labeling $\mathbf{q}^*$ such that $\mathbf{w}_{\text{sup}}$ minimizes the semi-supervised risk $R_\phi^{\text{semi}}(\mathbf{w}, \mathbf{X}, \mathbf{y}, \mathbf{X}_{\text{u}}, \mathbf{q}^*)$. This risk function is strictly convex because the supervised risk is strictly convex and therefore $\mathbf{w}_{\text{sup}}$ is its unique minimizer. This immediately implies that for every $\mathbf{w}_{\text{semi}} \neq \mathbf{w}_{\text{sup}}$, we have that $R_\phi^{\text{semi}}(\mathbf{w}_{\text{semi}}, \mathbf{X}, \mathbf{y}, \mathbf{X}_{\text{u}}, \mathbf{q}^*) > R_\phi^{\text{semi}}(\mathbf{w}_{\text{sup}}, \mathbf{X}, \mathbf{y}, \mathbf{X}_{\text{u}}, \mathbf{q}^*)$. $\qquad\square$

For decreasing margin-based losses, we now show that we can always explicitly construct a $\mathbf{q}^*$, such that the corresponding semi-supervised solution equals the original supervised one. With this, a result similar to Theorem 1 for the soft-assignment guarantee directly follows, but first we formulate that explicit construction of the necessary soft labeling.

**Lemma 2.** *If $\phi$ is a decreasing convex margin-based loss function where for each unlabeled object $\mathbf{x}$, the derivatives $\phi'(-\mathbf{x}^\top \mathbf{w}_{\mathrm{sup}})$ and $\phi'(\mathbf{x}^\top \mathbf{w}_{\mathrm{sup}})$ exist, we can recover $\mathbf{w}_{\mathrm{sup}}$ by minimizing the semi-supervised loss by assigning responsibilities $\mathbf{q} \in [0,1]^U$ as*

$$q = \frac{\phi'(-\mathbf{x}^\top \mathbf{w}_{\mathrm{sup}})}{\phi'(\mathbf{x}^\top \mathbf{w}_{\mathrm{sup}}) + \phi'(-\mathbf{x}^\top \mathbf{w}_{\mathrm{sup}})} \,, \tag{6}$$

*if $\phi'(\mathbf{x}^\top \mathbf{w}_{\mathrm{sup}}) + \phi'(-\mathbf{x}^\top \mathbf{w}_{\mathrm{sup}}) \neq 0$, and any $q \in [0,1]$ otherwise.*

*Proof.* Consider the case where we have one unlabeled object $\mathbf{x}$ with responsibility $q$. The semi-supervised objective then becomes

$$\begin{aligned} R_\phi^{\mathrm{semi}}(\mathbf{w}) = & R_\phi(\mathbf{w}, \mathbf{X}, \mathbf{y}) \\ & + q\phi(\mathbf{x}^\top \mathbf{w}) + (1-q)\phi(-\mathbf{x}^\top \mathbf{w}) \,. \end{aligned}$$

Since $\phi$ is convex, to guarantee that $\mathbf{w}_{\mathrm{sup}}$ is still a global minimizer of $R_\phi^{\mathrm{semi}}$, we need to find a $q \in [0,1]$ that causes the gradient of this objective, evaluated in $\mathbf{w}_{\mathrm{sup}}$, to remain equal to zero:

$$\begin{aligned} \nabla_\mathbf{w} R_\phi^{\mathrm{semi}}(\mathbf{w}_{\mathrm{sup}}) = & \mathbf{0} + q\phi'(\mathbf{x}^\top \mathbf{w}_{\mathrm{sup}})\mathbf{x} \\ & - (1-q)\phi'(-\mathbf{x}^\top \mathbf{w}_{\mathrm{sup}})\mathbf{x} \\ = & \mathbf{0} \end{aligned} \tag{7}$$

where $\phi'$ denotes the derivative of $\phi(a)$ with respect to $a$. As long as $\phi'(\mathbf{x}^\top \mathbf{w}_{\mathrm{sup}}) + \phi'(-\mathbf{x}^\top \mathbf{w}_{\mathrm{sup}}) \neq 0$, we can explicitly solve for $q$ to get

$$q = \frac{\phi'(-\mathbf{x}^\top \mathbf{w}_{\mathrm{sup}})}{\phi'(\mathbf{x}^\top \mathbf{w}_{\mathrm{sup}}) + \phi'(-\mathbf{x}^\top \mathbf{w}_{\mathrm{sup}})} \,. \tag{8}$$

If $\phi$ is a decreasing loss, then

$$\phi'(a) \leq 0$$

and for each object $0 \leq q \leq 1$. If $\phi'(\mathbf{x}^\top \mathbf{w}_{\mathrm{sup}}) + \phi'(-\mathbf{x}^\top \mathbf{w}_{\mathrm{sup}}) = 0$, because $\phi$ is decreasing, we know both $\phi'(\mathbf{x}^\top \mathbf{w}_{\mathrm{sup}}) = 0$ and $\phi'(-\mathbf{x}^\top \mathbf{w}_{\mathrm{sup}}) = 0$ and so any $q$ is allowed to satisfy (7), including $0 \leq q \leq 1$. Since $0 \leq q \leq 1$ for each object individually, we can do it for all objects to get a vector of responsibilities $\mathbf{q} \in [0,1]^U$. □

Now that we have shown by a constructive argument that for decreasing margin-based losses it always holds that $\mathbf{w}_{\mathrm{sup}} \in \mathcal{C}_\phi$, the following result is straightforward.

**Theorem 2.** *Let $\phi$ be a decreasing convex margin-based loss function and $\mathbf{w}_{\mathrm{sup}}$ be the unique minimizer of a strictly convex $R_\phi(\mathbf{w}, \mathbf{X}, \mathbf{y})$ and suppose for each unlabeled object $\mathbf{x}$, the derivatives $\phi'(-\mathbf{x}^\top \mathbf{w}_{\mathrm{sup}})$ and $\phi'(\mathbf{x}^\top \mathbf{w}_{\mathrm{sup}})$ exist. There is no semi-supervised classifier $\mathbf{w}_{\mathrm{semi}}$ for which Equation (5) holds, while having at least one $\mathbf{q}^*$ for which $D_\phi(\mathbf{w}_{\mathrm{semi}}, \mathbf{w}_{\mathrm{sup}}, \mathbf{X}, \mathbf{y}, \mathbf{X}_\mathrm{u}, \mathbf{q}^*) < 0$.*

*Proof.* This follows directly from Lemma 1 and Lemma 2. □

**Remark 2.** *The requirement to have a strictly convex supervised risk function can be relaxed. What we basically need in the proof is that $\mathbf{w}_{\mathrm{sup}}$ is the unique optimizer for $R_\phi^{\mathrm{semi}}(\mathbf{w}, \mathbf{X}, \mathbf{y}, \mathbf{X}_\mathrm{u}, \mathbf{q}^*)$. Nevertheless, unlike, for instance, a hinge loss that is not regularized by something like a 2-norm of the weight vector, many interesting objective functions are strictly convex.*

This result means that for decreasing loss functions it is impossible to construct a semi-supervised learner that is different from the supervised learner and, in terms of its surrogate loss on the full training data, is never outperformed by the supervised solution. In other words, if the semi-supervised classifier is taken to be different from the supervised classier, there is always the risk that there is a true labeling of the unlabeled data for which the loss of the semi-supervised learner on the full data becomes larger than the loss of the supervised one.

Is it unexpected that semi-supervised learning cannot be done safely in this strict setting? For whom it is not, it may then come as a surprise that there are margin-based losses for which it is actually possible to construct safe semi-supervised learners.

# 4 Possibilities for Safe SSL

If we look beyond the decreasing losses, and consider those that can increase as well, we may yet be able to get a classifier that is guaranteed to be better than the supervised solution in terms of the surrogate loss, even in the pessimistic regime. When can we expect safe semi-supervised learning to allow for improvements of its supervised counterpart? And if improvements are possible, how then do we construct an actual classifier that does so in a safe way?

To construct a semi-supervised learner that at least is guaranteed to never be worse, we need to find $\mathbf{w}_{\text{semi}}$, the $\mathbf{w}$ that minimizes $D_\phi(\mathbf{w}, \mathbf{w}_{\text{sup}}, \mathbf{X}, \mathbf{y}, \mathbf{X}_{\text{u}}, \mathbf{q})$ for all possible $\mathbf{q}$. This corresponds, more precisely, to the following minimax problem:

$$\min_{\mathbf{w}} \max_{\mathbf{q} \in [0,1]^U} D_\phi(\mathbf{w}, \mathbf{w}_{\text{sup}}, \mathbf{X}, \mathbf{y}, \mathbf{X}_{\text{u}}, \mathbf{q}). \tag{9}$$

This is a formulation similar to the one used by Loog (2016), where instead of margin-based losses, the loss functions are log-likelihoods of a generative model. It is clear that Equation (9) can never be larger than 0. This simply indicates that we can always find a semi-supervised learner that is at least as good as the supervised one, by simply sticking to the supervised solution. To show that we can do better than that, consider the following.

If $R_\phi^{\text{semi}}$ is convex in $\mathbf{w}$, then since the objective is linear in $\mathbf{q}$ and $[0,1]^U$ is a compact space we can invoke (Sion, 1958, Corrolary 3.3), which states that the value of the minimax problem is equal to the value of the maximin problem:

$$\max_{\mathbf{q} \in [0,1]^U} \min_{\mathbf{w}} D_\phi(\mathbf{w}, \mathbf{w}_{\text{sup}}, \mathbf{X}, \mathbf{y}, \mathbf{X}_{\text{u}}, \mathbf{q}). \tag{10}$$

Assume the function $D_\phi$ is strictly convex in $\mathbf{w}$ for every fixed $\mathbf{q}$. Now suppose $\mathbf{w}_{\text{sup}}$ is not in $\mathcal{C}_\phi$. In that case, the inner minimization in Equation (10) is always strictly smaller than 0 for each $\mathbf{q}$ because of the strict convexity of the loss. This means that Equation (10) is strictly smaller than 0 and in turn the same holds for Equation (9).

So, if $\mathbf{w}_{\text{sup}} \notin \mathcal{C}_\phi$, $\mathbf{w}_{\text{semi}}$ will strictly improve upon $\mathbf{w}_{\text{sup}}$.

## 4.1 Some Sufficient Conditions

So all that is required to show that the procedure just described leads to an improved classifier is therefore that $\mathbf{w}_{\text{sup}} \notin \mathcal{C}_\phi$. For an unlabeled data set consisting of a single sample $\mathbf{x}$, this is readily done by reconsidering the proof of Lemma 2 and the argument in the previous paragraph. In particular, rewriting Equation (7), we can conclude the following:

**Lemma 3.** *Let $\mathbf{X}_{\text{u}} = \mathbf{x}^\top$ and $\phi$ be a margin-based loss function where the derivatives $\phi'(-\mathbf{x}^\top \mathbf{w}_{\text{sup}})$ and $\phi'(\mathbf{x}^\top \mathbf{w}_{\text{sup}})$ exist and $R_\phi^{\text{semi}}$ be strictly convex. If there is no $q \in [0,1]$ such that*

$$(\phi'(\mathbf{x}^\top \mathbf{w}_{\text{sup}}) + \phi'(-\mathbf{x}^\top \mathbf{w}_{\text{sup}}))\mathbf{x}q = (\phi'(-\mathbf{x}^\top \mathbf{w}_{\text{sup}}))\mathbf{x}$$

*then $\mathbf{w}_{\text{sup}} \notin \mathcal{C}_\phi$ so $\mathbf{w}_{\text{semi}}$ has to be different from $\mathbf{w}_{\text{sup}}$ and, therefore, the former has to improve over the latter.*

The case in which $U > 1$ turns out to be hard to fully characterize. Again starting from Equation (7), we can state that if there is no $\mathbf{q} \in [0,1]^U$ such that

$$\sum_{i=1}^U q_i \phi'(\mathbf{x}_i^\top \mathbf{w}_{\text{sup}})\mathbf{x}_i - (1 - q_i)\phi'(-\mathbf{x}_i^\top \mathbf{w}_{\text{sup}})\mathbf{x}_i = \mathbf{0}$$

then the gradient evaluated in the supervised solution of the objective function over all training data is not zero and so the semi-supervised solution is different, therefore improving over the supervised solution. But this result is hardly insightful. For one, it is unclear if this at all happens when $U > 1$. We do, however, have a sufficient condition that leads the semi-supervised learner to improve over the supervised counterpart. For this, we consider convex, margin-based losses $\phi$ that are decreasing to the left of 1 and to the right of 1 start to increase, as for instance, in the cases of the quadratic or absolute loss. So these losses increasingly penalize overestimation of the label value of every object.

Table 1: Margin-based loss functions and their corresponding responsibilities

| **Name** | $\phi(y\mathbf{x}^\top\mathbf{w}_{\text{sup}})$ | $q(\mathbf{x}^\top\mathbf{w}_{\text{sup}})$ | **Range** |
|---|---|---|---|
| Logistic | $\sqrt{2}\log(1+\exp(-y\mathbf{x}^\top\mathbf{w}_{\text{sup}}))$ | $(1+\exp(-\mathbf{x}^\top\mathbf{w}_{\text{sup}}))^{-1}$ | $(0,1)$ |
| Hinge | $\max(1-y\mathbf{x}^\top\mathbf{w}_{\text{sup}},0)$ | $\begin{cases}\frac{1}{2}, & \text{if } -1 < \mathbf{x}^\top\mathbf{w}_{\text{sup}} < 1 \\ 1, & \text{if } \mathbf{x}^\top\mathbf{w}_{\text{sup}} > 1 \\ 0 & \text{if } \mathbf{x}^\top\mathbf{w}_{\text{sup}} < -1 \end{cases}$ | $\{0,\frac{1}{2},1\}$ |
| Exponential | $\exp(-y\mathbf{x}^\top\mathbf{w}_{\text{sup}})$ | $\frac{\exp(\mathbf{x}^\top\mathbf{w}_{\text{sup}})}{\exp(-\mathbf{x}^\top\mathbf{w}_{\text{sup}})+\exp(\mathbf{x}^\top\mathbf{w}_{\text{sup}})}$ | $(0,1)$ |
| Quadratic | $(1-y\mathbf{x}^\top\mathbf{w}_{\text{sup}})^2$ | $\frac{1}{2}(\mathbf{x}^\top\mathbf{w}_{\text{sup}}+1)$ | $(-\infty,\infty)$ |
| Absolute | $\|1-y\mathbf{x}^\top\mathbf{w}\|$ | $\begin{cases}\frac{1}{2}, & \text{if } -1 < y\mathbf{x}^\top\mathbf{w}_{\text{sup}} < 1 \\ \text{No solution}, & \text{otherwise} \end{cases}$ | $\{\frac{1}{2}\}$ |

**Theorem 3.** *Let*

$$\phi'(a)\begin{cases}\leq 0, & \text{if } a \leq 1 \\ > 0, & \text{if } a > 1,\end{cases}$$

*and $R_\phi^{\text{semi}}$ be strictly convex. If, for all $\mathbf{x} \in \mathbf{X}_{\text{u}}$, $|\mathbf{x}^\top\mathbf{w}_{\text{sup}}|$ is larger than 1, then $\mathbf{w}_{\text{semi}} \neq \mathbf{w}_{\text{sup}}$. That is, we get an improved semi-supervised estimator if all points in $\mathbf{X}_{\text{u}}$ are outside of the margin.*

The restriction that all points should be outside of the margin is, of course, rather strong. But, as indicated, the requirement is only sufficient and certainly not necessary. The proof, as well as an alternative condition for improvement for the quadratic loss are provided in the supplementary material.

## 5 Examples

Table 1 shows the implied responsibilities $q(\mathbf{x}^\top\mathbf{w}_{\text{sup}})$ for loss functions corresponding to a number of well-known classifiers. The table contains both examples of decreasing losses and losses that also strictly increase. In this first group, the range of the responsibilities will always be between $[0,1]$, meaning the (partial) labels of the unlabeled data can always be set in such a way that the supervised solution is obtained from the semi-supervised objective function. This in turn implies that no safe semi-supervised method exists for these losses. This shows, for instance, that it is not possible to construct a safe semi-supervised version of the support vector machine or for logistic regression. In the second case (for quadratic and absolute losses) it is not always possible to set the responsibilities in such a way as to recover the supervised solution and a safe semi-supervised classifier is sometimes possible.

A more thorough description of these examples, as well as a more precise characterization for when to expect improvements in case of the quadratic loss, is provided in the supplementary material.

## 6 Discussion

As Seeger (2001) and others have argued, for diagnostic methods, where $p(y|\mathbf{x})$ gets modeled directly and not through modeling the joint distribution $p(y,\mathbf{x})$, semi-supervised learning without additional assumptions should be impossible because the parameters of $p(y|\mathbf{x})$ and $p(\mathbf{x})$ are a priori independent. Considering why these methods do not allow for safe semi-supervised versions offers a different understanding of why this claim may or may not be true. While our results applied to logistic regression corroborates their claim, the quadratic loss shows a counterexample. This shows that for losses that strictly increase over some interval, even safe improvements can be possible in the diagnostic setting. One important strength of our analysis is that we also consider the minimization of loss functions that may not induce a correct probability. It is the decreasingness of the loss, rather than correspondence to a probabilistic model that determines whether safe semi-supervised learning is possible. Moreover, some of the losses for which safe semi-supervised learning is possible are successfully applied in supervised learning in practice and it is therefore interesting that safe semi-supervised versions exist.

Our results also might seem to contradict the result by Sokolovska et al. (2008) (and, by extension Kawakita & Takeuchi (2014)) that, when the supervised model is misspecified, a particular semi-supervised adaptation of logistic regression has an asymptotic variance that is at least as small as supervised logistic regression. In this work, however, we cover the pessimistic setting where a semi-supervised learner needs to outperform the supervised learner for all possible labelings in a finite sample setting. This is a much stricter requirement than the asymptotic result in (Sokolovska et al., 2008).

The (negative) result presented here is in line with the conclusions of Ben-David et al. (2008), who show that the worst-case sample complexity of a supervised learner is at most a constant factor higher than that of any semi-supervised approach for a classifier over the real line, and they conjecture this result holds in general. Darnstädt et al. (2013) prove that a slightly altered and more precise formulation of this conjecture holds when hypothesis classes have finite VC-dimension, while they show that it does not hold for more complex hypothesis classes. Whereas these works consider generalization bounds on the error rate in the PAC learning framework, in our work, we considered a more conservative or pessimistic setting of safe semi-supervised learning, while considering performance on a finite sample in terms of the surrogate loss. This leads to an alternative explanation why these (strict) improvements are not possible for some losses, similar to the claim in Ben-David et al. (2008). It also leads, however, to the contrasting conclusion that for some losses, these improvements are possible (even when the VC dimension is finite), which contradicts the claim of Ben-David et al. (2008) that improvements are not possible unless strong assumptions about the distribution of the labels are made.

The improvement guarantee, in terms of classification accuracy, of the safe semi-supervised SVM by Li & Zhou (2015) depends on the assumption that the true labeling of the objects is given by one of the low-density separators that their algorithm finds. In our analysis we avoid making such assumptions. The consequence of this is that all possible labelings have to be considered, not just those corresponding to a low-density separator. If their low-density assumptions holds, their method provides one way of making use of this information to guarantee safe improvements. As we have demonstrated, however, in a worst case sense no such guarantees can be given, at least in terms of the semi-supervised objective considered in our work. Without making these untestable assumptions, our results show a safe semi-supervised support vector machine is impossible.

For loss functions that are strictly increasing over some interval, safe improvement is possible. One could ascribe this fact to a peculiar property of these losses: they give increasingly higher loss even if the sign of the decision function is correct. The improvements in terms of the loss that we get may therefore not be useful for classification, since they may be in a part of the loss function where the surrogate loss already forms a bad approximation to the $\{0, 1\}$-loss. In the supervised case, however, surrogate losses like the quadratic loss generally give decent performance in terms of the error rate, e.g. competitive with SVMs (Rifkin et al., 2003). It is therefore not surprising either that its pessimistic semi-supervised counterpart has also shown increased performance (Krijthe & Loog, 2017a,b).

# 7 Conclusion

We have shown that for the class of convex margin-based losses, the fact whether they are decreasing or not plays a key role in whether they admit safe semi-supervised procedures. In particular, we have shown that, without making additional assumptions, it is impossible to construct safe semi-supervised procedures for decreasing losses by deriving what partial assignment of the unlabeled objects leads to the recovery of the supervised classifier from a semi-supervised objective. This subsequently implied that if we choose any semi-supervised procedure that deviates from the supervised solution, there is some labeling of the unlabeled objects (which could be the true labeling) for which it decreases performance. While this means that for many supervised procedures it is impossible to construct a safe semi-supervised learner in this strict sense, some losses do admit such solutions. A less strict guarantee might admit performance improvement by aiming for semi-supervised solutions that in expectation rather than on any particular dataset, outperform their supervised counterparts.

The stark reality is that if one sticks to strictly safe semi-supervised learning, besides opportunities for some surrogate losses, there are clear limits to the development of such procedures.

**Acknowledgements**

We thank Alexander Mey for his constructive feedback on an earlier version of this manuscript. This work was funded by Project P23 of the Dutch COMMIT research programme.

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
