[Supplementary Material · limits_pessimistic_ssl_supplementary.pdf]

# The Pessimistic Limits and Possibilities of Margin-based Losses in Semi-supervised Learning: Supplementary Material

**Proof: Theorem 3**

*Proof.* Without loss of generality, we can assume that we have translated, rotated, and scaled our data such that the supervised solution is given by $\mathbf{w}_{\text{sup}} = (1, 0, \ldots, 0)^\top$. Such standardization of the data does not lead to an essentially different problem.

It is easy to check that for every $\mathbf{x}_i^\top \mathbf{w}_{\text{sup}} = \mathbf{x}_{i1} > 1$, we have $\phi'(\mathbf{x}_i^\top \mathbf{w}_{\text{sup}}) = \phi'(\mathbf{x}_{i1}) > 0$, where $\mathbf{x}_{i1}$ indicates the first coordinate of sample $\mathbf{x}_i$. Likewise, we have $\phi'(-\mathbf{x}_i^\top \mathbf{w}_{\text{sup}}) = \phi'(-\mathbf{x}_{i1}) < 0$. It therefore follows, for every choice of $q_i \in [0, 1]$, that $q_i \phi'(\mathbf{x}_i^\top \mathbf{w}_{\text{sup}})\mathbf{x}_{i1} - (1 - q_i)\phi'(-\mathbf{x}_i^\top \mathbf{w}_{\text{sup}})\mathbf{x}_{i1} > 0$. Likewise, for every $\mathbf{x}_i^\top \mathbf{w}_{\text{sup}} = \mathbf{x}_{i1} < -1$, we have the same result: for every choice of $q_i \in [0, 1]$, $q_i \phi'(\mathbf{x}_i^\top \mathbf{w}_{\text{sup}})\mathbf{x}_{i1} - (1 - q_i)\phi'(-\mathbf{x}_i^\top \mathbf{w}_{\text{sup}})\mathbf{x}_{i1} > 0$. This shows that the first equation in the system given by

$$\sum_{i=1}^{U} q_i \phi'(\mathbf{x}_i^\top \mathbf{w}_{\text{sup}})\mathbf{x}_i - (1 - q_i)\phi'(-\mathbf{x}_i^\top \mathbf{w}_{\text{sup}})\mathbf{x}_i = \mathbf{0}$$

does not equal 0, and so the gradient differs from zero, meaning that the supervised solution cannot be the optimal one. $\square$

## Example: Logistic Loss

Consider the logistic loss function given by

$$\phi(y\mathbf{x}^\top \mathbf{w}) = \log(1 + \exp(-y\mathbf{x}^\top \mathbf{w})),$$

and whose minimization leads to the logistic regression classifier. Its derivative is given by

$$\phi'(y\mathbf{x}^\top \mathbf{w}) = \frac{-\exp(-y\mathbf{x}^\top \mathbf{w})}{1 + \exp(-y\mathbf{x}^\top \mathbf{w})},$$

from which we can verify it is a decreasing loss. Applying Equation (6) we find that

$$q = \frac{-\exp(\mathbf{x}^\top \mathbf{w}_{\text{sup}})}{1 + \exp(\mathbf{x}^\top \mathbf{w}_{\text{sup}})}$$
$$\times \left( \frac{-\exp(-\mathbf{x}^\top \mathbf{w}_{\text{sup}})}{1 + \exp(-\mathbf{x}^\top \mathbf{w}_{\text{sup}})} + \frac{-\exp(\mathbf{x}^\top \mathbf{w}_{\text{sup}})}{1 + \exp(\mathbf{x}^\top \mathbf{w}_{\text{sup}})} \right)^{-1}.$$

Because the second term equals $-1$, after rewriting the first term, we have

$$q = \frac{1}{1 + \exp(-\mathbf{x}^\top \mathbf{w}_{\text{sup}})}.$$

Thus we see that the responsibility assigned to the new object is exactly the class posterior assigned by logistic regression.

## Example: Support Vector Machine

The hinge loss, employed in support vector classification, has the form

$$\phi(y\mathbf{x}^\top \mathbf{w}) = \max(1 - y\mathbf{x}^\top \mathbf{w}, 0).$$

The value for the derivative at all points except $1 - y\mathbf{x}^\top \mathbf{w} = 0$ is given by

$$\phi'(y\mathbf{x}^\top \mathbf{w}) = \begin{cases} -1, & \text{if } 1 - y\mathbf{x}^\top \mathbf{w} > 0 \\ 0, & \text{otherwise}. \end{cases}$$

Plugging this into Equation (6), we have that

$$q = \begin{cases} \frac{1}{2}, & \text{if } -1 < \mathbf{x}^\top \mathbf{w}_{\text{sup}} < 1 \\ 1, & \text{if } \mathbf{x}^\top \mathbf{w}_{\text{sup}} > 1 \\ 0 & \text{if } \mathbf{x}^\top \mathbf{w}_{\text{sup}} < -1. \end{cases}$$

If the prediction is strongly positive (respectively, negative), it will be assigned to the positive (negative) class. If on the other hand, it is within the margin, it gets assigned to both classes equally. It means that for the unlabeled objects in the margin, any change in $\mathbf{x}^\top \mathbf{w}_{\text{sup}}$ has an opposite contribution for the part of the loss corresponding to the positive and the negative class. Only by weighting the two options equally will a change in $\mathbf{x}^\top \mathbf{w}_{\text{sup}}$ not yield a change in the gradient of the semi-supervised objective at the supervised solution.

**Example: Quadratic Loss**

Now consider the quadratic loss, also referred to as the squared loss, which is not a decreasing loss function:
$$\phi(y\mathbf{x}^\top \mathbf{w}) = (1 - y\mathbf{x}^\top \mathbf{w})^2 \,.$$
This non-decreasingness might make the quadratic loss seems less applicable as a surrogate loss for classification than decreasing losses. Quadratic loss is, however, a widely used and often discussed surrogate loss for classification (though it is not always explicitly identified as a surrogate quadratic loss). Some well-known works discussing some form of squared loss minimization for classification are (Hastie et al., 1994; Poggio & Smale, 2003; Suykens & Vandewalle, 1999). Rifkin et al. (2003), Fung & Mangasarian (2001) and, in a related setting, Rasmussen & Williams (2005, Sec.3.7) also provide proof of the good performance of the quadratic loss as compared to, for instance, standard support vector machines, while allowing for algorithms that are less computationally demanding. So in the supervised setting, the quadratic loss can be an effective surrogate loss. This leads to the question whether quadratic loss based classifiers generalized to the semi-supervised setting can guarantee non-degradation of the loss compared to the supervised classifier.

The derivative of the quadratic loss is
$$\phi'(y\mathbf{x}^\top \mathbf{w}) = -2(1 - y\mathbf{x}^\top \mathbf{w}) \,.$$
Again using (6) we find that
$$q = \frac{-2(1 + \mathbf{x}^\top \mathbf{w}_{\text{sup}})}{-2(1 - \mathbf{x}^\top \mathbf{w}_{\text{sup}}) - 2(1 + \mathbf{x}^\top \mathbf{w}_{\text{sup}})}$$
which we can simplify to
$$q = \frac{\mathbf{x}^\top \mathbf{w}_{\text{sup}} + 1}{2} \,.$$
This is a rescaling of the decision function from the interval $[-1, +1]$ to $[0, 1]$. Note that in this case $\mathbf{x}^\top \mathbf{w}$ is not necessarily restricted to be within $[-1, +1]$ and so it may occur that $q \notin [0, 1]$. In this case there is no assignment of the responsibilities that recovers the supervised solution and thus the unlabeled data forces us to update the decision function $\mathbf{x}^\top \mathbf{w}$ for the semi-supervised classifier.

When $U > 1$, for the decreasing loss functions, it was enough to show that each $q_i \in [0, 1]$ can be set individually in order to reconstruct the supervised solution using a responsibility vector $\mathbf{q} \in [0, 1]^U$. For the quadratic loss, however, the situation is more complex when multiple unlabeled objects are available. This is because, considering each $q_i$ individually might not allow us to find $\mathbf{q} \in [0, 1]^U$ for which the gradient of the semi-supervised objective at the supervised solution is equal to zero, but there could still be a combined $\mathbf{q} \in [0, 1]^U$ for which this does hold, as we discussed for the general case in Theorem 3.

It turns out that if the dimensionality $d \geq U$, such a $\mathbf{q} \in [0, 1]^U$ does not exist as long as
$$\mathbf{X}_{\text{u}} \mathbf{w}_{\text{sup}} \notin [-1, 1]^U \,.$$
If $d \leq U$, however, then it is guaranteed that such $\mathbf{q} \in [0, 1]^U$ does not exist if
$$\|\mathbf{X}_{\text{u}} \mathbf{w}_{\text{sup}}\|_2 > \sqrt{U} \,.$$
These results are essentially different and stronger than Theorem 3 as it shows that even if some of the unlabeled points are within the margin, the semi-supervised learner has to be different from the supervised learner if one or more of the unlabeled points are sufficiently far outside of the margin. The proof can be found below. While this proof is specific to the quadratic loss, perhaps a stricter version of the more generally applicable result in Theorem 3 is also possible.

**Example: Absolute Loss**

The absolute loss is given by
$$\phi(y\mathbf{x}^\top\mathbf{w}) = |1 - y\mathbf{x}^\top\mathbf{w}|.$$
and its derivative at all values except $1 - y\mathbf{x}^\top\mathbf{w} = 0$ then becomes
$$\phi'(y\mathbf{x}^\top\mathbf{w}) = \begin{cases} -1, & \text{if } 1 - y\mathbf{x}^\top\mathbf{w} > 0 \\ +1, & \text{otherwise}. \end{cases}$$

When $-1 < y\mathbf{x}^\top\mathbf{w}_{\text{sup}} < 1$, we can use Equation (6) to find $q = \frac{1}{2}$. Otherwise, $\phi'(\mathbf{x}^\top\mathbf{w}_{\text{sup}}) + \phi'(-\mathbf{x}^\top\mathbf{w}_{\text{sup}}) = 0$ but unlike for decreasing losses, now there is no $q$ that makes the gradient of the semi-supervised objective in the supervised solution equal to zero. In that case, when we have a single unlabeled object, the semi-supervised solution is an improvement over the supervised solution. For the case of multiple unlabeled objects it may again be possible to find a vector of responsibilities $\mathbf{q} \in [0,1]^U$ that recovers the supervised solution. Again, Theorem 3 offers a sufficient condition where the semi-supervised solution must improve over its supervised counterpart.

**Proof: Improvement for Quadratic Loss**

*Proof.* Whether it is possible to find some $\mathbf{q}$ for which minimizing the semi-supervised objective gives the supervised solution in case of the quadratic loss comes down to the question whether the system of equations
$$\mathbf{X}_{\text{u}}^\top\mathbf{q} = \frac{1}{2}(\mathbf{X}_{\text{u}}^\top\mathbf{1} + \mathbf{X}_{\text{u}}^\top\mathbf{X}_{\text{u}}\mathbf{w}_{\text{sup}}) \tag{11}$$
has a solution $\mathbf{q} \in [0,1]^U$. Let $(\mathbf{X}_{\text{u}}^\top)^+$ denote the Moore-Penrose pseudo-inverse of $\mathbf{X}_{\text{u}}^\top$. We consider two scenarios: $d \geq U$, the number of unlabeled objects is smaller or equal to the dimensionality of the feature vectors, and $d \leq U$, where we have more unlabeled objects than dimensions.

If $d \geq U$, the pseudo-inverse can be written as $(\mathbf{X}_{\text{u}}\mathbf{X}_{\text{u}}^\top)^{-1}\mathbf{X}_{\text{u}}$ meaning we have a unique solution
$$\mathbf{q} = \frac{1}{2}(\mathbf{1} + \mathbf{X}_{\text{u}}\mathbf{w}_{\text{sup}})$$
and so the supervised solution cannot be recovered unless $\mathbf{X}_{\text{u}}\mathbf{w}_{\text{sup}} \in [-1,1]^U$.

If $d \leq U$, the pseudo-inverse can be written as $(\mathbf{X}_{\text{u}}^\top)^+ = \mathbf{X}_{\text{u}}(\mathbf{X}_{\text{u}}^\top\mathbf{X}_{\text{u}})^{-1}$. Rewriting Equation (11) in terms of $\mathbf{r} = 2\mathbf{q} - 1$, the condition for improvement is that
$$\mathbf{X}_{\text{u}}^\top\mathbf{r} = \mathbf{X}_{\text{u}}^\top\mathbf{X}_{\text{u}}\mathbf{w}_{\text{sup}}$$
has no solution $\mathbf{r} \in [-1,1]^U$. Solving this using the pseudo-inverse we find the solution $\mathbf{r}^+$ with the smallest norm among all possible solutions:
$$\mathbf{r}^+ = \mathbf{X}_{\text{u}}\mathbf{w}_{\text{sup}}.$$
We therefore have for any solution $\mathbf{r}$ that
$$\|\mathbf{r}\|_2 \geq \|\mathbf{X}_{\text{u}}\mathbf{w}_{\text{sup}}\|_2$$
and so if $\|\mathbf{X}_{\text{u}}\mathbf{w}_{\text{sup}}\|_2 > \sqrt{U}$, this implies that every solution $\mathbf{r}$ lies outside of the unit cube $[-1,1]^U$ and no proper solution of responsibilities exists. $\square$