[Reviews · NeurIPS 2018]

Reviewer 1



The paper presents some impossibility results for semi-supervised learning with margin based losses that decrease with increasing margin (includes popular losses like hinge loss, log loss, etc). They are of the following form: take a classifier w_sup which is trained using only labeled data; take a semi-supervised classifier w_semi (\neq w_sup) that is learned using any semi-sup algorithm utilizing unlabeled data as well; consider a composite loss L(w) on labeled and unlabeled *training* examples; the results say that there exists a labeling of the unlabeled samples for which L(w_sup) is smaller than L(w_semi). It is shown that for margin based losses that increase in a region (such as least squares loss), such a labeling may not exist. Although the impossibility result is interesting, I am not convinced about its use or significance. The proofs are essentially based on constructing a labeling of unlabeled samples such that samples with higher margins on the positive side with w_semi are assigned -ve negative label (and vice versa). It is unlikely if impossibility results obtained with this type of post-hoc pessimistic labeling will have any bearing on real semi-supervised learning settings. ================== I have read the author response. However, I still have concerns on significance and usefulness of the results.

Reviewer 2



The paper develops an analysis of semi supervised classification in a transductive context for linear classifiers and convex losses. It shows that in most cases semi-supervised learning cannot be guaranteed to bring an improvement over supervised learning. This is a worst case analysis called “pessimistic” in the paper. This result is extended to soft labels. A characterization of possible situations where semi-supervised learning could be beneficial is also introduced. The paper is well motivated and clear. It is well illustrated by detailed examples in the appendix. My main concern is the worst case context of the derivations. The results are general, sure, but they are weak at the same time. The main results says that one cannot guarantee that a semi-supervised learner will not degrade the performance of a supervised learner for all possible target configuration of unlabeled data and at the same time have a better performance for at least one target configuration. As a worst case analysis, this does not take into consideration the nature of the problem, or the distribution of examples. This is then quite far from any “real” situation. The conditions obtained for the characterization of favorable conditions are weak too. Overall, the paper is correct and presents an interesting result, but this is a weak result. Update: I wouild like to thank the authors for the feedback, but there is no new argument there to change my opinion.

Reviewer 3



Overview and Recommendation: Many popular binary classifiers are defined by convex margin-based surrogate losses such as SVMs and Logistic regression. Designing a semi-supervised learning algorithm for these classifiers, that is guaranteed to improve upon the "lazy" approach of throwing away the unlabeled data and just using the labeled data while training, is of considerable interest, because of the time-consuming experimentation that the use of SSL currently requires. This paper analyzes this problem and the results presented in the paper are primarily of theoretical interest. I had great difficulty in rating the significance of this work, therefore my own confidence rating is only 3. The proofs of the theorems use elementary steps. I checked them in detail and they are correct, but, the significance of the theorems themselves was hard to measure. There is definitely some importance for theorem 3 as a good counter-example for common intuition and previous conjectures, therefore I have given the paper a rating of 6. Summary: The paper gives a new definition of a "safe" semi-supervised binary classifier as a classifier that is guaranteed to attain a lower loss on the combined labeled-unlabeled dataset than the loss achived by a supervised classifier learnt from just the labeled data. Under this definition of safety, theorems 1 and 2 in the paper prove that a safe semi-supervised classifier/hypothesis does not exist for monotonically decreasing convex margin-based, e.g. SVMs, and therefore safe SSL is impossible for such classifiers. Next, a more surprising result is shown in theorem 3, that it is possible to do safe SSL for some special convex surrogate loss functions that are not monotonically decreasing, e.g. the quadratic loss. Pros: - This gives a very general proof for the common intuition that it is not possible to do semi-supervised learning for discriminative classifiers without making assumptions about the joint distribution of inputs and labels. - One important strength of our analysis is that we also consider the minimization of loss functions that may not induce a correct probability. - There are no assumptions on the input features so the inputs can be the features extracted from the last layer of a neural network and the results will continue to hold. Cons: - In Theorems 1 and 2, the paper shows that "safe" semi-supervised learning, under an extremely stringent definition of safety, is impossible to perform for "reasonable" convex surrogate loss functions such as hinge loss and logistic loss, which seems unsurprising. And on the other side the paper proves that "safe" semi-supervised learning is possible for some "unreasonable" classification surrogate loss functions such as the squared-loss. My main concern is that it is not clear how useful "safe" semi-supervised learning on squared surrogate loss function is since it is not a common surrogate loss anyway. The authors do anticipate this objection in line 304-307 but they really need to give many more examples for their claim that "surrogate losses like the quadratic loss generally give decent performance". - The definition of safety in this work is very stringent and fairly non-standard. For example, (Sakai 2017) present estimation error bounds for PNU learning and they present sufficient condition when learning from positive-negative and unlabeled data is likely to outperform supervised learning with high probability. The authors should give more motivation for why their definition of safety is preferable compared to Sakai et al. [Sakai 2017] "Semi-Supervised Classification Based on Classification from Positive and Unlabeled Data" by Sakai et al. ICML, (2017) === After Author Response === Based on the additional references for the use of squared loss provided by authors in their response, I feel better about the use of squared error loss. I will encourage the authors to make space for discussing these references, even if briefly, in the main body of the paper since it can make the audience more accepting of the ideas in the paper. Note that the other reviewers had similar doubt about the usefulness of the results.